# Comprehensive Analysis of the *PP2C* Gene Family in Grape (*Vitis vinifera* L.) and Identification of *VvPP2C26* and *VvPP2C41* as Negative Regulators of Fruit Ripening

**DOI:** 10.3390/plants14243827

**Published:** 2025-12-16

**Authors:** Kaidi Li, Kai Liu, Keyi Wang, Yunning Pang, Xuzhe Zhang, Xiujie Li, Bo Li

**Affiliations:** 1Shandong Academy of Grape, Shandong Academy of Agricultural Sciences, Jinan 250100, China; lkd965322@163.com (K.L.); liukai2429@163.com (K.L.); pangyunning@163.com (Y.P.); lixiujie-2007@163.com (X.L.); 2College of Agricultural Science and Technology, Shandong Agriculture and Engineering University, Jinan 250100, China; caaswangkeyi@163.com; 3College of Life Sciences, Shandong Agricultural University, Taian 271000, China; zhangxuzhe2002@163.com

**Keywords:** grape, *PP2C* gene family, fruit ripening, abscisic acid, *VvPP2C26*, *VvPP2C41*

## Abstract

Protein phosphatase 2Cs (PP2Cs) are members of the serine/threonine phosphatase family that play pivotal roles in regulating plant development and responses to environmental stresses. However, comprehensive genome-wide studies of the *PP2C* gene family in grape (*Vitis vinifera* L.) have not yet been conducted. In the present study, 78 *VvPP2C* genes were identified and classified into 12 clades based on their phylogenetic relationships. Analysis of physicochemical properties and gene/protein architectures revealed that the members within each clade shared conserved structural features. Synteny analysis demonstrated that both tandem and segmental duplications substantially contributed to the expansion of the *VvPP2C* gene family. Tissue-specific transcriptional profiles and cis-element analyses indicated the potential involvement of these genes in grape development and stress responses. Moreover, expression analysis identified *VvPP2C26* and *VvPP2C41* as the most abscisic acid (ABA)-responsive genes, with expression patterns highly correlated with grape berry development. Functional validation in transgenic tomato lines demonstrated that the overexpression of either gene markedly delayed fruit ripening. Collectively, this study provides new insights into the evolutionary diversification and regulatory functions of the *PP2C* gene family in grape and identifies *VvPP2C26 and VvPP2C41* as key candidates for elucidating ABA-mediated ripening mechanisms in non-climacteric fruits.

## 1. Introduction

Protein phosphatase 2C (PP2C) is a metal-dependent serine/threonine phosphatase belonging to the protein phosphatase M (PPM) family. It serves as a key regulator of diverse cellular processes, including environmental stress responses [1], hormonal signaling [2], and development [3]. PP2Cs possess a conserved catalytic core of approximately 300 amino acids containing 11 invariant motifs essential for phosphatase activity [1]. Although this catalytic domain is typically located at the C-terminus, certain isoforms exhibit N-terminal positioning [4]. Structural analyses have indicated that the flexible surface loops within the catalytic core mediate specific protein–protein interactions, conferring functional diversity among PP2C members [5]. In *Arabidopsis thaliana*, the PP2C family comprises more than 80 members that are phylogenetically grouped into 12 clades (A–K) [6,7]. Notably, clade A PP2Cs (PP2CAs) function as central negative regulators of abscisic acid (ABA) signaling [6,8]. The core ABA signaling module is highly conserved: upon hormone binding, PYR/PYL/RCAR receptors undergo conformational changes that inhibit PP2CA activity [9,10]. This inhibition releases SnRK2 kinases from repression, allowing their phosphorylation and activation, often mediated by upstream Raf-like kinases [11]. Activated SnRK2s subsequently phosphorylate downstream targets, such as ABF transcription factors and SLAC1 ion channels, thereby initiating ABA-dependent physiological responses [12].

As one of the most widely cultivated fruit crops globally, grape (*Vitis vinifera* L.) holds high economic value and a long history of domestication [13,14]. However, maintaining an optimal balance between fruit quality, nutritional development, and reduced postharvest perishability remains a major challenge [15,16,17]. More than 30% of harvested fruits, including grapes, are lost annually due to postharvest spoilage [18]. Such losses not only cause economic waste but also exacerbate nutritional insecurity, particularly in regions where fruits are primary dietary sources of vitamins, antioxidants, and fiber [18,19]. Enhancing postharvest durability through molecular breeding (e.g., gene editing and epigenome engineering) has become a key strategy for developing cultivars that satisfy both agronomic and consumer demands under changing climatic conditions. Fruit ripening is controlled by a complex regulatory network that integrates hormonal signaling, transcriptional reprogramming, and epigenetic modification [20,21,22]. In climacteric fruits such as tomato [23] and apple [24], ethylene (ETH) is the principal regulator of ripening, whereas in non-climacteric fruits such as grape [25] and strawberry [26], ABA serves as the dominant hormonal signal. In climacteric fruits, ETH promotes autocatalytic biosynthesis through ACC synthase and ACC oxidase (ACS/ACO) and activates major transcription factors, including RIN (MADS-box), NOR (NAC), and ERFs [24,27]. Conversely, non-climacteric fruit ripening is largely governed by ABA, whereas the molecular mechanisms and signaling components underlying ABA-mediated regulation remain less well defined than those of ethylene. This knowledge gap limits the precise genetic manipulation of ripening grapes and restricts the development of varieties with improved storage potential and desirable sensory traits.

Extensive research has established PP2Cs as critical regulators of stress and immune signaling, fine-tuning adaptive responses through interactions with SnRK2 kinases, transcription factors, and ion channels [28]. For instance, in *Triticum aestivum*, TaPP2C158 dephosphorylates TaSnRK1.1, thereby suppressing the drought tolerance [29]. In *Arabidopsis*, the calcium sensor SCaBP8 inactivates PP2C.D6 and PP2C.D7 to activate the SOS1 Na^+^/H^+^ exchanger under salt stress [30]. Similarly, AP2C1 negatively regulates immune responses by suppressing ethylene biosynthesis and defense-related gene expression during pathogen attacks [31]. Recent studies have revealed the involvement of PP2Cs in fruit ripening, particularly in tomatoes. Silencing *SlPP2C1* accelerates ripening, elevates ABA accumulation, and enhances ABA sensitivity, whereas it also results in defects in flower development [32]. Similarly, SlPP2C2 dephosphorylates the transcriptional repressor SlZAT5, stabilizing its inhibitory function on ripening-related genes and delaying ethylene-mediated ripening [33]. Another phosphatase, SlPP2C3, suppresses ripening by interacting with ABA receptors and SlSnRK2.8, whereas its downregulation promotes earlier ripening and increases fruit glossiness through altered cuticular metabolism [34]. Despite these advances in tomato, the functions and regulatory mechanisms of PP2Cs in grape berry ripening remain largely unknown.

Given the pivotal role of ABA in non-climacteric fruit ripening and the established involvement of PP2Cs in ABA-mediated signaling, PP2C proteins are key modulators of grape berry ripening. Although the *PP2C* gene families have been comprehensively characterized in several species, including *A. thaliana* [30], cucumber (*Cucumis sativus*) [35], strawberries (*Fragaria vesca* and *Fragaria ananassa*) [36], peanut (*Arachis hypogaea*) [37], wheat (*Triticum aestivum*) [38], and soybean (*Glycine max*) [39], a systematic genome-wide analysis in grape remains unavailable. In this study, we identified 78 *VvPP2C* genes from the grapevine genome and systematically analyzed their physicochemical properties, gene structures, chromosomal distributions, evolutionary relationships, and expression profiles across multiple tissues. Furthermore, we investigated the expression dynamics of *VvPP2CAs* during berry development and under exogenous ABA treatment. Functional characterization of transgenic tomato lines further elucidated the roles of *VvPP2C26 and VvPP2C41* in regulating fruit ripening. These findings provide new insights into the molecular functions of VvPP2Cs and facilitate future molecular breeding efforts aimed at improving fruit quality and postharvest performance.

## 2. Results

### 2.1. Identification of VvPP2Cs

To identify *PP2C* genes in the grapevine genome (PN_T2T), a BLASTP search was performed using *Arabidopsis* PP2C protein sequences as queries. The initial screening yielded 634 candidate proteins. Subsequent validation using the NCBI-CDD database confirmed the presence of complete PP2C domains in 78 proteins, which were designated VvPP2C1 to VvPP2C78 according to their chromosomal positions and linear order (Appendix A). Analysis of physicochemical properties revealed substantial variation among the encoded proteins: protein lengths ranged from 92 (VvPP2C31) to 1083 (VvPP2C49) amino acids; molecular weights (MW) varied from 9.77 kDa (VvPP2C31) to 120.06 kDa (VvPP2C49); and predicted isoelectric points (pI) ranged from 4.31 (VvPP2C54) to 9.40 (VvPP2C32). All VvPP2C proteins were predicted to localize to intracellular compartments, primarily the nucleus, chloroplasts, cytoplasm, and plasma membrane, although several were also predicted to reside in the endoplasmic reticulum and mitochondria (Table 1).

### 2.2. Phylogenetic Classification and Syntenic Relationships of VvPP2Cs

The evolutionary relationships among the 78 VvPP2C proteins were analyzed using a Neighbor-Joining (NJ) tree approach (Figure 1A). Following the classification scheme established for *Arabidopsis*, in which the *PP2C* genes were grouped into 13 clades [30], the *VvPP2C* genes were assigned to 12 clades, as no homologs corresponding to *Arabidopsis* clade G were identified in the grapevine genome. Eight *VvPP2Cs* (*VvPP2C10*/*23*/*24*/*35*/*54*/*65*/*66*/*74*) could not be assigned to any known clade. Clade A contained the largest number of members (11), whereas clade I contained only one. The *VvPP2C* genes were unevenly distributed across the 18 chromosomes (Figure 1B). Chromosome 18 harbored the greatest number (9), whereas chromosome 15 lacked *VvPP2C* genes entirely. The expansion of the *VvPP2C* family was further investigated by examining gene duplication events. Among all *VvPP2C* genes, 19 (24.35%) underwent 10 tandem duplication events, and 28 (35.89%) formed 14 segmental duplication pairs (Figure 1B; Appendix A). These results indicated that both tandem and segmental duplications contributed substantially to the expansion of the *VvPP2C* family. Synteny analysis revealed 71 and 81 orthologous gene pairs between *V. vinifera* and *A. thaliana* and between *V. vinifera* and *S. lycopersicum*, respectively, implying that grapevine shares a closer evolutionary relationship with tomato than with *Arabidopsis* (Figure 1C; Appendix A).

### 2.3. Analysis of Protein/Gene Structures of VvPP2Cs

To evaluate the structural variation and evolutionary relationships among *VvPP2C* genes, motif analysis was performed using MEME, and the gene structures were examined using GSDS 2.0. An unrooted phylogenetic tree constructed from protein sequences supported the clade classification previously established (Figure 1A and Figure 2A). Ten conserved motifs, ranging from 15 to 49 amino acids in length, were identified (Figure 2B and Appendix A). The members within each clade exhibited largely consistent motif compositions and arrangements. Motif 2 was universally present across all clades, whereas Motifs 1, 3, 4, 6, 7, and 9 were broadly distributed across clades A–I. In contrast, Motif 5 was specific to clades C and D, and Motif 8 was unique to clade D.

The exon–intron architectures of the 78 *VvPP2C* genes were also analyzed, revealing exon counts ranging from 1 to 26. *VvPP2C26* contained the highest number of exons (26), whereas *VvPP2C12*/*31*/*50*/*59*/*62*/*74*/*75* contained only one. Most members of clades B to E possessed both 5′- and 3′-UTRs, whereas these regions were detected in only a few genes belonging to clades A, F1, F2, H, I, J, and K (Figure 2C). Such variations in gene architecture suggest potential functional divergence among *VvPP2C* members.

### 2.4. Analysis of Promoters of VvPP2Cs

To explore the transcriptional regulation and potential biological roles of *VvPP2C* genes, the promoter regions were analyzed using PlantCARE. Five categories of cis-acting elements were identified, including five hormone-responsive elements, five stress-related elements, and three development-associated elements. Among these, ABA-responsive elements were the most prevalent (Figure 3A). The composition and abundance of cis-elements varied markedly among clades. Clade D harbored the highest number of development-related elements, Clade E was enriched in the stress-responsive elements, and Clade A contained the greatest proportion of hormone-responsive elements (Figure 3B). These differences in cis-element distribution exhibited functional diversification among *VvPP2C* clades, enabling them to mediate responses to hormonal, environmental, and developmental cues.

### 2.5. Tissue-Specific Expression Profiles of VvPP2Cs

To further elucidate the potential functions of *VvPP2C* genes, their expression patterns were examined across 21 grapevine organs and tissues at various developmental stages using data from the BAR database (Figure 4). Overall, the *VvPP2C* genes were constitutively expressed in nearly all examined tissues, although their expression levels varied substantially among tissues and developmental stages. These genes were grouped into five distinct expression clusters (Groups I–V). The genes in Groups I and II showed the highest expression across all tissues, whereas those in Groups III and IV exhibited consistently low transcript levels. The expression of Group IV genes was particularly elevated in the roots and fruits relative to that in other tissues. Further analysis revealed divergent expression profiles among *VvPP2C* clades and even among members within the same clade. For instance, in Group A involving 11 genes (*VvPP2C4*/*5*/*13*/*26*/*41*/*42*/*43*/*47*/*48*/*60*/*64*), *VvPP2C13*/*26*/*41* exhibited markedly higher expression across multiple tissues than other members.

### 2.6. Expression Changes in Vvpp2cas in Grape Berries in Response to Exogenous ABA

In model plants, clade A PP2Cs are recognized as key regulators of the ABA signaling pathway [40]. To investigate the responsiveness of grape *VvPP2CAs* to exogenous ABA, the expression patterns were analyzed following ABA treatment (Figure 5). Grape berries at the E–L 39 stage were treated with exogenous ABA and maintained at 23 °C. The expression levels of *VvPP2CAs* were examined at six subsequent time points. Most *VvPP2CAs* were transiently upregulated by ABA, with expression peaking at either 2 or 6 h post-treatment (HPT), followed by a gradual decline. This rapid induction-repression trend was particularly evident for *VvPP2C4*/*5*/*13*/*26*/*41*/*42*/*43*/*60*. In particular, *VvPP2C13*/*26*/*41* exhibited a sharp expression decrease after 6 HPT. In contrast, *VvPP2C47*/*48* also exhibited transient induction at 6 HPT but lacked consistent expression dynamics. In contrast, *VvPP2C64* exhibited a steady decrease in expression throughout the treatment period, without a distinct peak. Notably, *VvPP2C26 and VvPP2C41* displayed the most prominent transcriptional alterations in response to ABA, highlighting their potential roles as pivotal regulators within the ABA-mediated signaling network.

### 2.7. Expression Profiles of VvPP2CAs During Grape Berry Development

Given the enrichment of hormone- and development-related cis-elements in *VvPP2CA* promoters and their sensitivity to exogenous ABA, we further examined the expression profiles of these genes during grape berry development, with total soluble solids (TSS) serving as a physiological indicator of maturation (Figure 6A and Appendix A). Transcript levels of most *VvPP2CAs* changed dynamically across development stages. *VvPP2C4*/*5*/*13*/*26*/*41*/*48* were significantly downregulated, while *VvPP2C47* was upregulated. In contrast, *VvPP2C42*/*43*/*60*/*64* exhibited fluctuating expression without a clear trend. Notably, the downregulation of *VvPP2C26* and *VvPP2C41* was the most pronounced (Figure 6B). Pearson correlation analysis further revealed their expression levels were significantly and inversely correlated with TSS (for *VvPP2C26*, r = −0.879, *p* < 0.01; for *VvPP2C41*, r = −0.965, *p* < 0.01) (Appendix A), underscoring a close association between these genes and the progression of grape berry ripening.

### 2.8. VvPP2C26 and VvPP2C41 Acts as a Negative Regulator of Fruit Ripening

Given the pronounced ABA responsiveness and developmental expression changes in VvPP2C26 and *VvPP2C41*, these genes were hypothesized to function as negative regulators of fruit ripening. To validate their roles, overexpression (OE) lines of *VvPP2C26* and *VvPP2C41* were generated in tomato via *Agrobacterium*-mediated transformation, along with empty vector (EV) controls. qPCR verification confirmed the successful overexpression of target genes in six independent transgenic lines for each construct (Figure 7A). The lines *VvPP2C26*-OE#2 and *VvPP2C41*-OE#4 exhibiting the highest expression levels were selected for phenotypic assessment. Both *VvPP2C26*-OE#2 and *VvPP2C41*-OE#4 fruits displayed a marked delay in ripening compared with EV controls (Figure 7B). These results demonstrate that *VvPP2C26* and *VvPP2C41* function as negative regulators of fruit ripening, likely by modulating ABA-mediated signaling pathways.

## 3. Discussion

The PP2C family of protein phosphatases plays a central role in coordinating plant developmental processes and adaptive responses to environmental stresses [8,41]. Although the genome-wide analyses of the *PP2C* gene family have been performed in several plant species, including *A. thaliana* [30], cucumber (*C. sativus*) [35], Strawberries (*F. vesca* and *F. ananassa*) [36], peanut (*A. hypogaea*) [37], Wheat (*T. aestivum*) [38], and soybean (*G. max*) [39], a comprehensive investigation in grapevine had not been conducted prior to the present study. A total of 78 *VvPP2C* genes were identified through a genome-wide screening and systematically analyzed for their physicochemical characteristics, gene structures, chromosomal distributions, and expression dynamics during fruit development and in response to ABA treatment. In particular, *VvPP2C26* and *VvPP2C41* were examined in depth to elucidate their potential functions in grape berry development and ripening. These results establish a foundation for future functional characterization of the *VvPP2C* gene family in grape.

The *PP2C* family exhibits considerable evolutionary diversity and a long history of adaptation in land plants, with its functional divergence likely reflecting environmental adaptability [42]. In this study, 78 *VvPP2C* genes were identified and classified into 12 clades based on phylogenetic analysis and sequence alignment (Appendix A). Comparative genomic investigation revealed that the *VvPP2C* family in grapes was smaller than those in *A. thaliana* [28], *A. hypogaea* [37], and *T. aestivum* (Appendix A). This difference may reflect the influence of genome size and pseudogenization events during evolution. Grapevine with a genome size of approximately 500 Mbp contained fewer *PP2C* genes than *A*. *thaliana* (~135 Mbp) (Appendix A), likely due to the higher proportion of noncoding DNA in eukaryotic genomes [43]. Phylogenetic analysis further confirmed the classification of *VvPP2Cs* into 12 clades corresponding to the 13 clades defined in *Arabidopsis*, with no homologs identified for clade G and eight unclassified *VvPP2Cs* (Figure 1A). These observations highlight the influence of species-specific evolution on *PP2C* gene architecture [44]. Clade A of VvPP2Cs displayed greater heterogeneity than clade I (Figure 1A,B), suggesting an earlier evolutionary origin and longer time span for gene duplication and structural diversification [45]. Both tandem and segmental duplication events were major drivers of *VvPP2C* family expansion, accounting for 24.35% and 35.89% of the genes, respectively (Figure 1B; Appendix A). These duplication events likely contributed to the adaptive evolution of grapevine PP2Cs. Moreover, synteny analysis demonstrated more orthologous gene pairs between grapevine and tomato than between grape and *Arabidopsis* (Figure 1C; Appendix A), implying a closer evolutionary relationship between these two fruit crops [46]. This highlights the value of our study in providing a direct comparator for deciphering gene family evolution in fleshy fruits, as opposed to the more distant reference of *Arabidopsis*.

Structural analyses revealed that the conserved motifs of VvPP2Cs could provide valuable insights into their evolutionary conservation and functional specialization [47]. The composition and exon-intron structures of VvPP2Cs were largely conserved within clades but diverged across them (Figure 2 and Appendix A). Motif 2 was universally present, whereas Motifs 5 and 8 were specific to clades C and D, respectively. The members of clades B to E typically contained both 5′- and 3′-UTRs, which were less common in other clades. Given that proteins with similar structures in *Arabidopsis* often exhibit analogous regulatory functions [48], these conserved structural characteristics suggest that VvPP2C members within the same clade may possess comparable biological roles. Promoter cis-element analysis revealed a clade-specific distribution of hormonal, stress-responsive, and development-associated elements (Figure 3). Strikingly, the promoter landscape of grapevine VvPP2CAs showed a distinct enrichment profile compared to their well-characterized *Arabidopsis* counterparts. While both are rich in ABA-responsive elements, the grapevine promoters harbored a more diverse array of elements linked to light, circadian control, and fruit development, suggesting their regulation may be integrated with a broader set of environmental and developmental cues specific to a perennial vine undergoing seasonal growth and fruit production. Tissue-specific expression profiling can provide essential predictions for the functional differentiation of gene families [49]. Expression analysis revealed that *VvPP2Cs* were constitutively expressed in nearly all examined tissues, whereas their transcript levels varied considerably, forming five distinct expression clusters (Figure 4). Such ubiquitous yet differential expression implies that VvPP2Cs play diverse but fundamental roles in grape growth and development. Notably, even within a single clade, such as clade A, certain members (*VvPP2C13*/*26*/*41*) exhibited markedly higher expression levels, suggesting functional divergence following gene duplication events.

The roles of *PP2C* genes in regulating stress and hormonal signaling have been well established in model plants [50,51,52]. PP2Cs modulate the ABA signaling and physiological adaptation by interacting with key proteins such as SnRK2s, ABA receptors, transcription factors, and ion channels [53,54,55]. Recent studies have highlighted their participation in fruit development [32,33,34], where climacteric and non-climacteric fruits are governed by distinct hormonal regulators, including ETH and ABA. In *Arabidopsis*, clade A PP2Cs function as central negative regulators of ABA signaling [56]. In the absence of ABA, PP2CAs (e.g., ABI1, ABI2, and HAB1) dephosphorylate key serine residues within the activation loops of SnRK2 kinases, thereby suppressing their activities [8]. Upon ABA binding, PYR/PYL/RCAR receptors interact with and inhibit PP2CAs, consequently releasing SnRK2s to transduce ABA signals [57]. Li et al. identified an “ABA–FaPYR1–FaPP2C–FaSnRK2” signaling module as a core regulatory mechanism controlling strawberry ripening and proposed a model for ABA-mediated regulation of non-climacteric fruit ripening [58]. Guided by these findings, we analyzed the expression patterns of clade *VvPP2CAs* during grape berry development and under exogenous ABA treatment (Figure 5, Figure 6 and Appendix A). *VvPP2C26 and VvPP2C41* exhibited the most prominent upregulation in response to ABA, and their expression levels were strongly positively correlated with fruit ripening progression. To further validate their functional roles, transgenic tomato lines overexpressing *VvPP2C26 and VvPP2C41* were generated. Both the *VvPP2C26*-OE and *VvPP2C41*-OE lines displayed a significant delay in fruit ripening (Figure 7), indicating that these genes act as negative regulators of fruit ripening through ABA-mediated signaling pathways.

## 4. Materials and Methods

### 4.1. Plant Material

Six-year-old ‘Kyoho’ grapevines were cultivated in the open-field experimental vineyard of the Shandong Academy of Grape in Tai’an, China (34.41° N, 112.46° E). Grape berries were collected at key developmental stages according to the E-L system [59], including young fruit (E-31, May 23), green fruit (E-32, June 6; E-33, June 20; E-34, June 30), veraison (E-35, July 10; E-36, July 20), and mature stages (E-37, July 30; E-38, August 4; E-39, August 9). Samples were obtained from three independent vines at each stage. Two bunches of uniform, disease-free grape berries of consistent size were harvested from the middle portion of each vine. Grape berries of uniform size were harvested from the middle portion of each cluster and pooled. At the E-39 stage, fruits exhibiting consistent growth and free from mechanical injury, pest infestation, or disease symptoms were selected from three independent vines. The grape berries were immersed in a 1 mM ABA solution for 15 min and subsequently incubated in a controlled environment at 23 °C under constant humidity. Samples were collected at 0, 2, 6, 12, 24, and 48 HPT, with three biological replicates per time point, using the 0 HPT samples as controls. Immediately after sampling, the berry flesh (pulp) was separated from the seeds and skin, rapidly frozen in liquid nitrogen, and stored at −80 °C until subsequent analyses.

### 4.2. Identification of VvPP2Cs

The protein sequences of *Arabidopsis* PP2Cs were retrieved from the TAIR database (https://www.arabidopsis.org/browse/gene_family, accessed on 17 March 2025) [60] and used as reference queries to search the *Vitis vinifera* PN_T2T genome [61] within the Winberige database (http://www.winberige.cc/ftp.html, accessed on 17 March 2025) using BLASTP v2.12.0+. Sequences with significant similarity (E-value < 1 × 10^−10^) were retained for further evaluation [62]. The presence of conserved PP2C domains was confirmed using the NCBI Conserved Domain Database (https://www.ncbi.nlm.nih.gov/Structure/cdd/wrpsb.cgi, accessed on 17 March 2025). The predicted *VvPP2C* proteins were analyzed for biochemical parameters, including MW and pI, using ExPASy ProtParam (https://web.expasy.org/protparam/, accessed on 18 March 2025). Subcellular localization was predicted using WoLF PSORT (https://wolfpsort.hgc.jp/, accessed on 18 March 2025) [63].

### 4.3. Phylogenetic and Synteny Analyses

Multiple sequence alignment was performed using ClustalW implemented in MEGA version 12. Phylogenetic trees were constructed using the NJ method with 1000 bootstrap replicates [64]. Gene duplication patterns within the *VvPP2C* family were analyzed using MCScanX [65]. Chromosomal locations were retrieved from the Winberige database, and collinearity and duplication relationships were visualized using TBtools v2.0 software [66].

### 4.4. Analysis of Gene Structure and Conserved Motifs

The exon–intron structures of *VvPP2C* genes were determined and visualized using GSDS (http://gsds.cbi.pku.edu.cn/, accessed on 10 April 2025) [67]. Conserved motifs were identified using the MEME suite (https://meme-suite.org/meme/tools/meme, accessed on 10 April 2025) [68]. Integrating motif composition and gene structure information, a comprehensive visualization was generated using TBtools v2.0 [66].

### 4.5. Promoter Analysis

Promoter sequences extending 2000 bp upstream of the translation initiation codon for each *VvPP2C* gene were extracted using TBtools v2.0. Putative cis-acting regulatory elements were identified using the PlantCARE database (http://bioinformatics.psb.ugent.be/webtools/plantcare/html/, accessed on 18 April 2025) [69]. The detected elements were classified according to their function, and their distributions were visualized using TBtools v2.0 [66].

### 4.6. Tissue-Specific Expression Profile Analysis

The expression profiles of *VvPP2C* genes across 21 grapevine tissues were obtained from the BAR database (https://bar.utoronto.ca/efp_grape/cgi-bin/efpWeb.cgi, accessed on 10 May 2025), which provides genome-wide expression data for major grape organs and tissues, including roots, stems, leaves, buds, flowers, fruits, seedlings, and pollen [70]. The *Vitis vinifera* v2.1 genome was used here [71]. The expression levels (FPKM values) were quality checked and log_2_-transformed for normalization. Heatmaps were generated using TBtools v2.0 to visualize the expression patterns of *VvPP2C* genes across tissues.

### 4.7. qRT-PCR Validation

The primers used for qRT-PCR were designed using SnapGene v1.1.3 software (Appendix A). Total RNA was extracted from the frozen powder of grape berry flesh using the RNAprep Pure Plant Plus Kit (TIANGEN, Beijing, China) following the manufacturer’s instructions, which included steps for cell lysis, homogenate processing, and purification. First-strand cDNA was synthesized using PrimeScript™ RT Master Mix (Takara, Shiga, Japan), following the manufacturer’s instructions. qRT-PCR amplification was performed using a LightCycler 480 System (Roche Diagnostics, Shanghai, China; headquartered in Basel, Switzerland) and TB Green Premix Ex Taq (Takara). *VvUBI* was used as an internal reference [69]. Relative expression levels were calculated using the 2^−ΔΔCT^ method [72].

### 4.8. Vector Construction and Plant Transformation

The CDS of *VvPP2C26* (Vitvi009987) and *VvPP2C41* (Vitvi015923) was amplified from cDNA synthesized from ‘Kyoho’ grapevine leaves and validated by sequencing. The verified CDS fragments were ligated into the plant expression vector pBWA(V)HS using the ABclonal MultiF Seamless Assembly Mix (ABclonal Technology, Wuhan, China). The EV and recombinant constructs pBWA(V)HS–*VvPP2C26* and pBWA(V)HS–*VvPP2C41* were introduced into *A. tumefaciens* strain GV3101 using the freeze–thaw transformation method. Tomato (*S. lycopersicum* cv. Ailsa Craig) plants were subsequently generated through *A. tumefaciens*-mediated transformation with 35S::*VvPP2C26* and 35S::*VvPP2C41* constructs [73]. The corresponding primer sequences are presented in Appendix A.

## 5. Conclusions

In this study, a total of 78 *PP2C* genes were identified and classified into 12 distinct clades in grapevine. Gene duplication events, including both segmental and tandem types, played a major role in the expansion of the *VvPP2C* gene family. Analyses of cis-regulatory elements and tissue-specific expression patterns suggested the potential involvement of *VvPP2Cs* in grape berries growth and stress adaptation. Clade A members (*VvPP2CAs*) exhibited marked differential expression during fruit development and in response to ABA treatment. Functional characterization further demonstrated that the overexpression of *VvPP2C26 and VvPP2C41* in tomato significantly delayed fruit ripening. Collectively, these findings provide new insights into the biological functions of *PP2C* genes in grape and underscore their potential application in the genetic modulation of fruit ripening traits.

## Figures and Tables

**Figure 1 plants-14-03827-f001:**
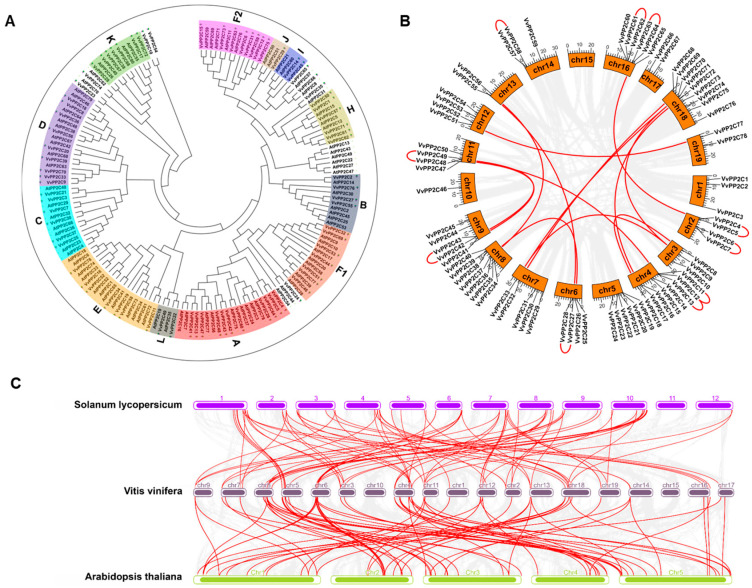
Phylogeny and genomic duplications of PP2Cs. (**A**) Phylogenetic relationships between PP2C proteins from *Vitis vinifera* and *Arabidopsis*, illustrated as a phylogenetic tree. Grape proteins are represented by purple circles, and *Arabidopsis* proteins by green triangles. (**B**) Chromosomal distribution and duplication events of *VvPP2Cs*. Red lines indicate genes derived from segmental or tandem duplications. (**C**) Synteny analysis of *VvPP2Cs* compared with *PP2Cs* in *Arabidopsis* and *Solanum lycopersicum*. Orthologous gene pairs are connected by red lines.

**Figure 2 plants-14-03827-f002:**
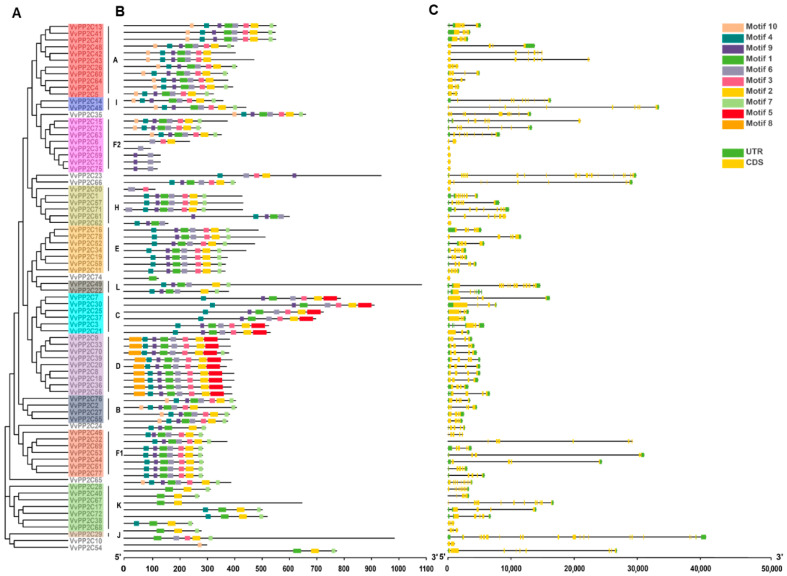
Phylogeny, conserved motifs, and exon-intron organization of VvPP2Cs. (**A**) Evolutionary relationships among VvPP2C proteins, as depicted in a phylogenetic tree. (**B**) Conserved motifs identified in the VvPP2C family. Ten distinct motifs are shown as colored boxes, with blue, red, and green representing motifs 1–10. (**C**) Exon-intron structures of *VvPP2C* genes. CDSs are represented by yellow boxes, UTRs by green boxes, and introns by black lines connecting them.

**Figure 3 plants-14-03827-f003:**
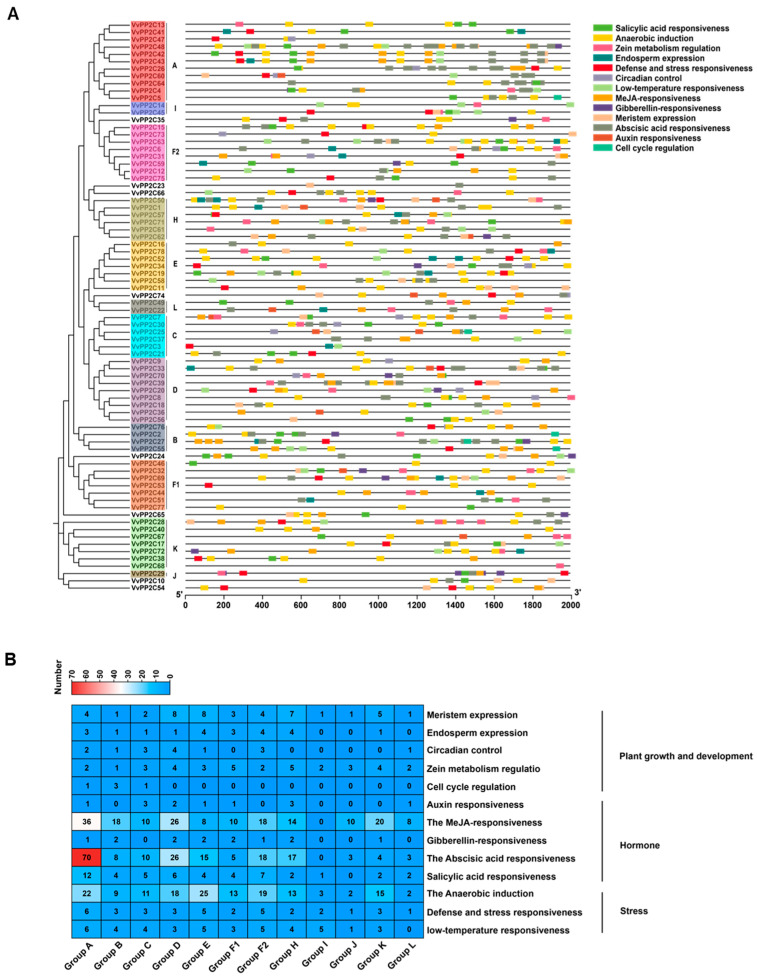
Composition and distribution of promoter cis-elements in *VvPP2Cs*. (**A**) Chromosomal locations of different cis-elements in *VvPP2C* promoters. Different colors indicate distinct types of cis-elements. (**B**) Heatmap showing the abundance of each cis-element type across promoters. The red and blue squares represent higher and lower counts, respectively.

**Figure 4 plants-14-03827-f004:**
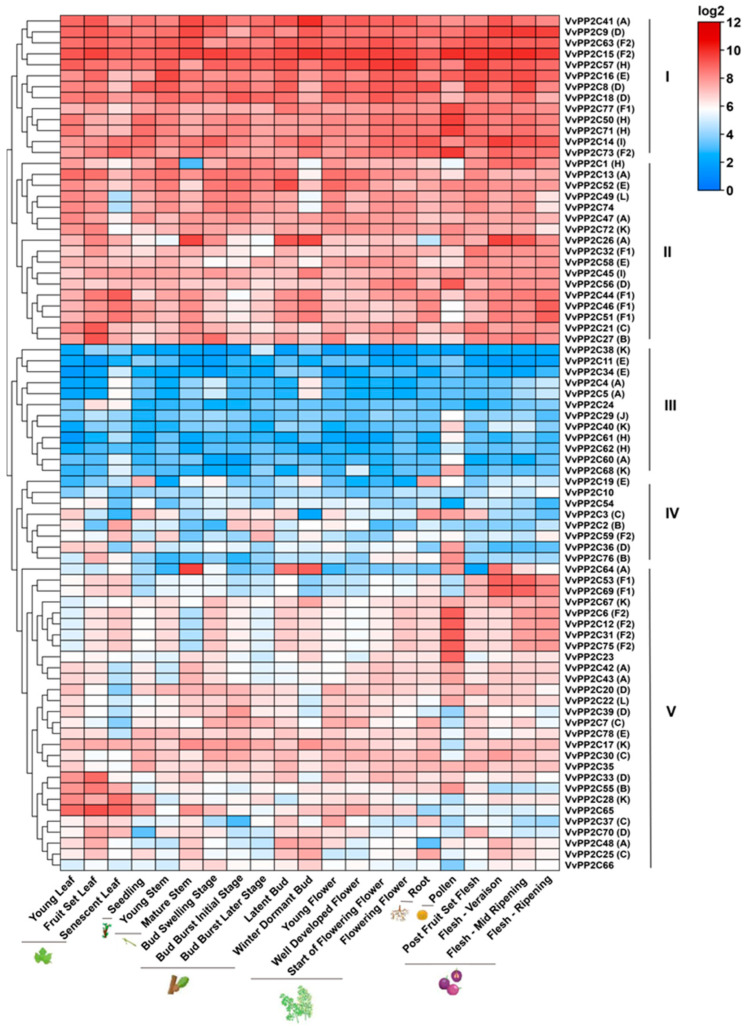
Tissue-specific expression profiles of *VvPP2Cs*. The heatmap was generated based on an in silico analysis of tissue-specific expression data retrieved from the BAR database. Expression values were log_2_-transformed and normalized, followed by hierarchical clustering. Differences in expression levels are indicated by a color gradient ranging from blue (low expression) to red (high expression).

**Figure 5 plants-14-03827-f005:**
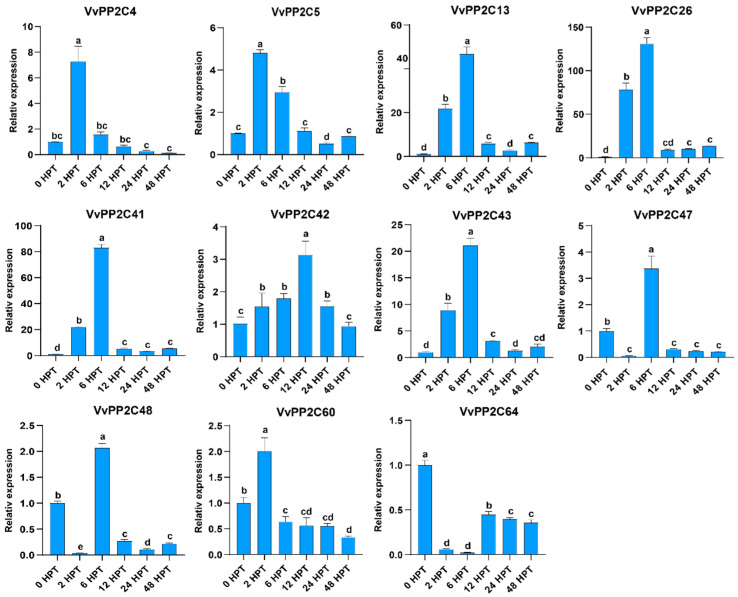
Expression profiles of *VvPP2CAs* in response to exogenous ABA treatment. Error bars represent SD (*n* = 3). Significant differences at *p* < 0.05, as assessed using Duncan’s multiple range test, are denoted by different lowercase letters.

**Figure 6 plants-14-03827-f006:**
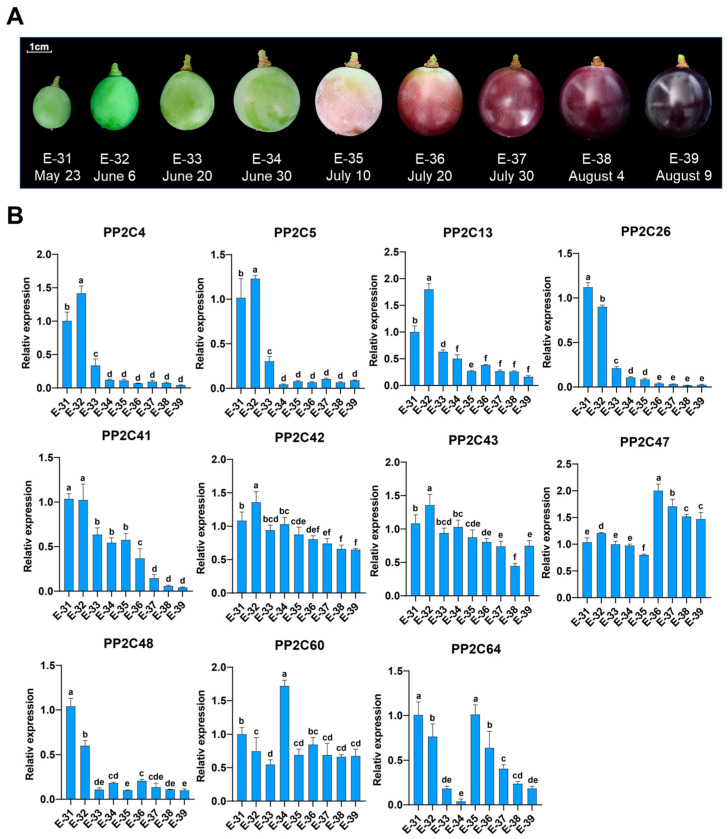
Expression profiles of *VvPP2CAs* during grape berry development. (**A**) Phenotypes of ‘Kyoho’ berries at different developmental stages. Sample names and collection dates are indicated below. Scale bar = 1 cm; (**B**) Expression patterns of *VvPP2CAs* during grape berry development. Error bars represent SD (*n* = 3). Significant differences at *p* < 0.05, as assessed using Duncan’s multiple range test, are denoted by different lowercase letters.

**Figure 7 plants-14-03827-f007:**
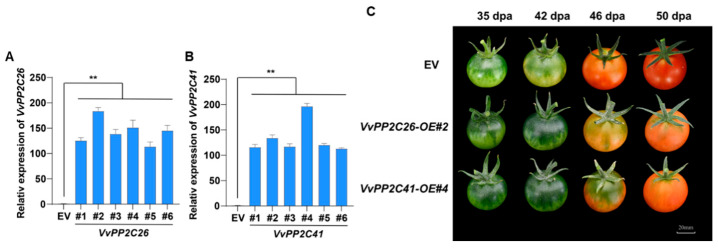
Overexpression of *VvPP2C26 and VvPP2C41* delays fruit ripening in tomato. (**A**,**B**) Relative expression levels of *VvPP2C26* (**A**) and *VvPP2C41* (**B**) in transgenic OE lines and EV controls. Error bars represent SD (*n* = 3). Asterisks denote significant differences (* *p* < 0.05; ** *p* < 0.01; Student’s *t*-test). (**C**) Representative phenotypes of EV control and transgenic tomato fruits at different developmental stages. DPA, days post-anthesis. Scale bar = 20 mm.

**Table 1 plants-14-03827-t001:** Properties of VvPP2Cs.

Name	ID	Chromosomal Location	Exon	Protein Length (aa)	MW (kDa)	pI	Subcellular Localization
VvPP2C1	Vitvi000366	chr1:3523203–3527912	8	427	46.07	8.52	nucl
VvPP2C2	Vitvi000748	chr1:7203836–7208412	5	406	44.49	8.56	nucl
VvPP2C3	Vitvi001964	chr1:27622889–27628604	3	524	57.62	5.05	nucl
VvPP2C4	Vitvi002118	chr2:1155304–1157053	3	393	42.72	5.53	nucl
VvPP2C5	Vitvi002137	chr2:1305671–1307156	3	322	35.27	6.28	cyto
VvPP2C6	Vitvi002873	chr2:8425242–8426525	2	235	25.48	5.11	cyto
VvPP2C7	Vitvi002887	chr2:8504870–8520986	4	787	86.79	5.22	nucl
VvPP2C8	Vitvi003776	chr3:1866552–1871660	4	397	43.77	8.20	nucl
VvPP2C9	Vitvi004360	chr3:6814270–6818089	4	381	42.45	8.98	mito
VvPP2C10	Vitvi004829	chr3:15381534–15382571	2	298	33.57	6.46	nucl
VvPP2C11	Vitvi005263	chr4:668828–670575	5	365	39.74	5.61	cysk
VvPP2C12	Vitvi005282	chr4:823128–823514	1	128	13.37	9.14	er
VvPP2C13	Vitvi005328	chr4:1184698–1189888	4	551	59.96	4.66	plas
VvPP2C14	Vitvi005885	chr4:6692565–6708865	10	357	39.15	5.01	cyto
VvPP2C15	Vitvi005948	chr4:7635262–7656256	9	373	40.34	5.35	nucl
VvPP2C16	Vitvi006510	chr4:18486517–18491801	5	486	53.65	5.84	nucl
VvPP2C17	Vitvi006587	chr4:19352268–19366244	4	501	53.24	6.51	mito
VvPP2C18	Vitvi006865	chr4:22011343–22016119	4	397	44.09	8.44	plas
VvPP2C19	Vitvi007259	chr5:491247–494246	5	373	41.26	5.79	nucl
VvPP2C20	Vitvi007363	chr5:1269792–1274884	4	370	41.05	6.60	plas
VvPP2C21	Vitvi007599	chr5:3371693–3375088	3	530	58.05	4.91	plas
VvPP2C22	Vitvi007695	chr5:4262748–4268128	3	379	41.71	4.74	plas
VvPP2C23	Vitvi007754	chr5:4879485–4909320	24	935	104.18	6.31	plas
VvPP2C24	Vitvi008101	chr5:8713684–8716370	5	294	32.99	9.24	mito
VvPP2C25	Vitvi009886	chr6:5601321–5604585	4	724	80.39	5.43	nucl
VvPP2C26	Vitvi009987	chr6:6343393–6344972	26	408	44.72	5.38	nucl
VvPP2C27	Vitvi010137	chr6:7586110–7588658	4	381	41.30	7.09	nucl
VvPP2C28	Vitvi010200	chr6:8147017–8150311	11	311	33.30	5.27	nucl
VvPP2C29	Vitvi011251	chr7:921238–962156	16	984	111.68	5.87	cyto
VvPP2C30	Vitvi011766	chr7:5125892–5133565	4	910	101.33	5.81	nucl
VvPP2C31	Vitvi011924	chr7:7116388–7116666	1	92	9.77	6.54	nucl
VvPP2C32	Vitvi012924	chr7:23506834–23536081	9	372	40.48	9.40	cyto
VvPP2C33	Vitvi013239	chr7:26376812–26380974	4	384	42.83	6.98	plas
VvPP2C34	Vitvi014354	chr8:10951216–10954053	5	441	48.58	7.09	nucl
VvPP2C35	Vitvi014501	chr8:12876468–12889600	12	659	72.99	5.50	mito
VvPP2C36	Vitvi014734	chr8:15133334–15136539	4	387	43.08	9.04	nucl
VvPP2C37	Vitvi014800	chr8:15793139–15795928	4	696	78.06	5.86	nucl
VvPP2C38	Vitvi015411	chr8:20939264–20940206	3	246	27.07	5.22	cyto
VvPP2C39	Vitvi015475	chr8:21467694–21472795	4	390	43.73	7.24	plas
VvPP2C40	Vitvi015683	chr8:23257885–23261197	5	270	29.77	9.20	mito
VvPP2C41	Vitvi015923	chr9:1668056–1671559	4	548	59.16	4.81	plas
VvPP2C42	Vitvi016080	chr9:3200757–3215695	7	402	43.61	5.39	nucl
VvPP2C43	Vitvi016084	chr9:3246328–3268743	10	470	51.75	5.75	plas
VvPP2C44	Vitvi016891	chr9:14273929–14298336	5	285	31.33	7.67	cyto
VvPP2C45	Vitvi017104	chr9:19144805–19178244	10	441	48.90	5.20	plas
VvPP2C46	Vitvi018841	chr10:14781765–14784116	5	283	31.85	5.91	cyto
VvPP2C47	Vitvi019711	chr11:1423634–1426793	4	550	59.78	4.93	plas
VvPP2C48	Vitvi019859	chr11:2535745–2549496	4	396	43.33	4.95	plas
VvPP2C49	Vitvi019885	chr11:2774861–2789507	15	1083	120.06	5.09	plas
VvPP2C50	Vitvi020255	chr11:6702481–6702810	1	109	11.89	4.89	cyto
VvPP2C51	Vitvi021422	chr12:4080558–4083605	5	283	31.09	6.87	cyto
VvPP2C52	Vitvi021684	chr12:6732006–6737729	5	473	52.60	5.69	nucl
VvPP2C53	Vitvi022127	chr12:11886584–11917676	5	282	31.19	7.76	mito
VvPP2C54	Vitvi022306	chr12:15144409–15171166	9	772	84.06	4.31	mito
VvPP2C55	Vitvi023077	chr13:734938–737231	4	374	40.38	6.67	mito
VvPP2C56	Vitvi023240	chr13:1948365–1954997	4	390	43.14	6.84	plas
VvPP2C57	Vitvi025444	chr14:1344752–1352871	7	430	46.27	7.97	nucl
VvPP2C58	Vitvi025487	chr14:1717965–1722450	5	366	40.76	6.43	nucl
VvPP2C59	Vitvi026444	chr14:15747910–15748296	1	128	13.60	8.93	plas
VvPP2C60	Vitvi030306	chr16:19502196–19507185	4	374	40.55	4.89	plas
VvPP2C61	Vitvi030367	chr16:20228603–20237757	7	600	67.47	6.59	mito
VvPP2C62	Vitvi030371	chr16:20247442–20247912	1	156	17.61	5.09	cyto
VvPP2C63	Vitvi030810	chr16:24494638–24502840	8	351	38.87	5.42	mito
VvPP2C64	Vitvi030821	chr16:24649661–24652358	4	375	40.83	6.58	plas
VvPP2C65	Vitvi031086	chr16:27313120–27316955	11	386	42.39	5.20	mito
VvPP2C66	Vitvi031427	chr17:3334742–3363981	10	402	44.12	6.65	mito
VvPP2C67	Vitvi031933	chr17:8100463–8117178	13	646	74.52	6.21	mito
VvPP2C68	Vitvi032843	chr18:1883860–1885417	4	279	30.93	5.93	cyto
VvPP2C69	Vitvi033094	chr18:4441846–4445569	5	283	30.92	7.05	mito
VvPP2C70	Vitvi033420	chr18:7847341–7851896	4	378	42.14	6.28	mito
VvPP2C71	Vitvi033554	chr18:8980478–8990160	9	429	46.33	5.26	nucl
VvPP2C72	Vitvi033704	chr18:10297572–10304337	5	519	55.99	7.31	mito
VvPP2C73	Vitvi033989	chr18:13009202–13022511	8	276	30.04	5.35	nucl
VvPP2C74	Vitvi034326	chr18:18775692–18776057	1	121	13.70	8.93	nucl
VvPP2C75	Vitvi034538	chr18:22941892–22942245	1	117	12.13	7.75	nucl
VvPP2C76	Vitvi035505	chr18:34932759–34936260	5	403	44.73	5.60	plas
VvPP2C77	Vitvi035925	chr19:2490155–2495950	5	283	31.10	6.35	cyto
VvPP2C78	Vitvi036699	chr19:10771058–10782615	5	511	55.38	5.20	nucl

## Data Availability

Data are contained within the article and Appendix A.

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
