# Peer review of "Comprehensive Analysis of the PP2C Gene Family in Grape (Vitis vinifera L.) and Identification of VvPP2C26 and VvPP2C41 as Negative Regulators of Fruit Ripening"

_plants, 2025, doi:10.3390/plants14243827_

Round 1
Reviewer 1 Report
Comments and Suggestions for Authors
This manuscript presents a comprehensive analysis of the grapevine PP2C gene family and functionally identifies VvPP2C26 and VvPP2C41 as negative regulators of fruit ripening. The study is valuable and is suitable for publication after minor revision to address the points below.
Major Comments:
- The manuscript requires thorough English editing for fluency and academic tone. Simplify overly long sentences and revise the wording such as “VvPP2C26/41” to “VvPP2C26 and VvPP2C41” in the title and keywords.
- Ensure consistency in figure citations; for instance, correct “VvRALFs” in the Figure 1C legend to “VvPP2Cs”.
- Unify terminology throughout: use one term (e.g., “HPT”) for post-treatment timepoints and standardize abbreviations for subcellular localizations (e.g., Nuc, Cyt, PM).
- The Discussion should more explicitly highlight the novelty of this study compared to prior work in model species like Arabidopsis and tomato.
- Clarify whether a mock control (e.g., solvent-only) was included in the ABA treatment experiment and state this in the Methods or figure legend.
- In all figures, specify what the error bars represent (SD or SEM) and state the number of biological replicates (n) for quantitative experiments.
- Specify the source of the “Vitvi” gene identifiers (e.g., PN_T2T genome) in the Methods or a footnote to aid reproducibility.
- Please cite more recent important grapevine studies.
Reviewer 2 Report
Comments and Suggestions for Authors
In the submitted manuscript by Bo Li and colleagues entitled “Comprehensive Analysis of the PP2C Gene Family in Grape and Identification of VvPP2C26/41 as Negative Regulators of Fruit Ripening”, the authors identified 78 PP2C genes (VvPP2Cs) across the grape genome and grouped into 12 distinct clades based on phylogenetic analysis. and RT-qPCR analysis. In addition, overexpression of VvPP2C26 or VvPP2C41 in transgenic tomato significantly delayed fruit ripening. Among the positive aspects of the manuscript are 1) the depth of bioinformatic analysis, 2) a well-defined goal, 3) a well-performed data integration to focus on the aim of the manuscript, 4) the introduction section is explanatory, and 5) in most cases, the statistical approach is clear. The negative aspects include: 1) the physiological data of berries is missing during fruit development, and 2) the correlation analysis is also missing. The manuscript is well-written; the figures have a good presentation. However, some issues should be addressed before the manuscript is accepted for publication.
Line 167: You mention ‘CDSs are represented by green boxes, UTRs by red boxes’. It is an obvious mistake. So, replace it with yellow and green as depicted in figure above.
Name the color scales in figures 3 and 4 as number and log2 (??), respectively.
In figures 1-3, you divided PP2C based on clades A, B, etc. This separation should be also kept in figure 4 added the letters after ‘VvPP2C4’.
Lines 232-234: The authors mention ‘In contrast, VvPP2C42/43/60/64 exhibited varying expression levels without a clear pattern. Among them, VvPP2C26/41 displayed the most distinct differential expression, which was strongly correlated with berry maturation (Figure 6B).’ Based on this choice, the following question arises. How was the maturation of the fruits determined by color, sugars or something else? It is typically stated that only the 2 genes are strongly correlated, however, to write this you must proceed to a Pearson correlation with some maturation characteristic as mentioned (color, SSC or whatever) which is currently lacking. Moreover, physiological data of berries during maturation should be included at least as supplementary material.
Lines 342-348: As there is a corresponding section (line 420), it should be deleted.
MM section
Lines 364-365: The authors mention ‘Immediately after sampling, all materials were rapidly frozen in liquid nitrogen and stored at −80°C until subsequent analyses.’ They need to clarify whether the sampling includes, for example, whole fruits, removal of pits and collection of peel and flesh of fruits or only flesh of fruits.
Reviewer 3 Report
Comments and Suggestions for Authors
Comments for the manuscript entitled "Comprehensive Analysis of the PP2C Gene Family in Grape and Identification of VvPP2C26/41 as Negative Regulators of Fruit Ripening" by Kaidi Li et al.
This study focuses on a less studied aspect, namely The ABA - mediated regulation of grape ripening, since the ripening of grapes - as non-climacteric fruits - is governed by ABA. This detailed investigation could contribute to the molecular improvement of grapevine varieties.
78 VvPP2C genes (classified into 12 distinct clades) were identified in the Vitis vinifera L. genome, and their physicochemical properties, genetic structures, chromosomal distributions, evolutionary relationships, and expression profiles in multiple tissues, as well as the dynamics of VvPP2CA expression during grape berries development and during exogenous ABA treatment. Synteny analysis revealed that Vitis vinifera has a closer evolutionary relationship with tomatoes than with Arabidopsis.
The results obtained highlight the role of PP2C genes in grapevine, suggesting their possible application in controlling grape ripening.
My comments is bellow:
- The title should also include the name of the species Vitis vinifera L.: .....grape (Vitis vinifera L.).
- Everywhere in the paper, where you wrote "berry", "berries" you should also add grape, grapes. That is: grape berries. Otherwise, they might be confused with berries (ex. blueberries, blackberries etc.)
- In line 93, it should be strawberries, not Strawberries. In line 94, it should be wheat, not Wheat, and in parentheses write the full Latin name, i.e.Triticum aestivum.
- In lines 96, 107, 125 you wrote"grape genome". The correct one is grapevine genome.
- In lines 211, 362, 365 it should be 23 0C, -80 0C respectively (at a certain distance).
- In line 396 you wrote "grape tissues". It should be grapevine tissues.
- You do not specify anything concrete about the procedures that followed after freezing the grape berries in different stages. At least the important steps. Ex. obtaining cell homogenates, ultracentrifudation, etc.
- In line 412 you wrote "grape leaves". It should be grapevine leaves.
- In lines 140 and 141, "grape" should be replaced with Vitis vinifera.
- In line 189 a you wrote "21 grape organs and tissues". Correct is 21 grapevine organs and tissues.
- In line 267, instead of grape, grapevine is more appropriate. This is because in that sentence you listed the names of species (not the fruits of these species) on which genome analyses were performed at the PP2C gene family level.
- In lines 282, 293, 294, 342, 347, 422 it should be grapevine, instead of grape or grapes.
- In lines 343, 425 it should be grape berries.
I wish the authours much success in publishing this manuscript!
Round 2
Reviewer 2 Report
Comments and Suggestions for Authors
All my concerns have been successfully addressed. Therefore I recommend the current manuscript for publication in Plants in its current form.